# Biosecurity and Mitigation Strategies to Control Swine Viruses in Feed Ingredients and Complete Feeds

**DOI:** 10.3390/ani13142375

**Published:** 2023-07-21

**Authors:** Gerald C. Shurson, Pedro E. Urriola, Declan C. Schroeder

**Affiliations:** 1Department of Animal Science, University of Minnesota, St. Paul, MN 55108, USA; urrio001@umn.edu; 2Department of Veterinary Population Medicine, University of Minnesota, St. Paul, MN 55108, USA; dcschroe@umn.edu

**Keywords:** biosecurity, chemical mitigants, feed, swine, thermal processing, virus detection methods

## Abstract

**Simple Summary:**

Global trade of feed ingredients that may be contaminated with significant concentrations of swine viruses is a concern for the potential transmission of swine diseases because viable virus particles can survive in feed ingredients and complete feed for several weeks or months. No global swine virus surveillance and monitoring system exists to determine the possible presence and concentrations of swine viruses in feed ingredients. Biosecurity protocols based on hazard analysis and risk-based preventive controls must be developed and implemented in feed ingredient supply chains to prevent virus contamination. In addition, mitigation strategies including the use of extended storage time, thermal and irradiation processing, and certain feed additives have been shown to provide partial virus inactivation in contaminated ingredients and complete feeds under specific conditions. However, analytical methods capable of accurately determining viable virus concentrations that can lead to infection are lacking and need to be developed. Effective feed mill decontamination strategies are also needed for various swine viruses. Several functional ingredients and nutrients such as spray-dried animal plasma, medium-chain fatty acids, and soy isoflavones have antiviral properties and have been shown to alleviate adverse health of pigs undergoing a viral disease challenge when included in diets.

**Abstract:**

No system nor standardized analytical procedures at commercial laboratories exist to facilitate and accurately measure potential viable virus contamination in feed ingredients and complete feeds globally. As a result, there is high uncertainty of the extent of swine virus contamination in global feed supply chains. Many knowledge gaps need to be addressed to improve our ability to prevent virus contamination and transmission in swine feed. This review summarizes the current state of knowledge involving: (1) the need for biosecurity protocols to identify production, processing, storage, and transportation conditions that may cause virus contamination of feed ingredients and complete feed; (2) challenges of measuring virus inactivation; (3) virus survival in feed ingredients during transportation and storage; (4) minimum infectious doses; (5) differences between using a food safety objective versus a performance objective as potential approaches for risk assessment in swine feed; (6) swine virus inactivation from thermal and irradiation processes, and chemical mitigants in feed ingredients and complete feed; (7) efficacy of virus decontamination strategies in feed mills; (8) benefits of functional ingredients, nutrients, and commercial feed additives in pig diets during a viral health challenge; and (9) considerations for improved risk assessment models of virus contamination in feed supply chains.

## 1. Introduction

Concerns about the transboundary transmission of swine viruses through international trade and travel [1] have led to increased interest in the role of feed as a potential virus transmission route. However, many transmission routes have been identified as having greater frequency of occurrence for swine virus transmission than feed [2,3,4,5,6]. Historically, bacterial, parasite, prion, and virus contamination of animal by-products and uncooked or inadequately heat-processed food waste have been associated with causing various types of animal disease, which led to the development and implementation of effective thermal and chemical mitigation strategies as part of quality control and feed safety programs in the feed industry [7]. However, it was not until the porcine epidemic diarrhea virus (PEDV) epidemic occurred in 2013 in North America that the potential for virus contamination of feed ingredients was more extensively promoted as a possible threat for disease transmission [8]. More recently, concerns about the transboundary transmission of swine viruses through international trade and travel [1] have led to considerable research to evaluate the effectiveness of extended storage times, thermal and irradiation processes, and chemical mitigants to inactivate swine viruses, as well as evaluation of decontamination strategies in feed mills as components of biosecurity programs of global feed supply chains.

The major swine viruses of concern for potential transmission through global feed supply chains are PEDV, African swine fever virus (ASFV), classical swine fever virus (CSFV), porcine respiratory and reproductive syndrome virus (PRRSV), Seneca Valley A virus (SVV-A), and foot and mouth disease virus (FMDV). Although a few reports have provided evidence that contamination of some viruses such as ASFV [9], CSFV [10], and SVV-A [11,12] has occurred in feed and feed ingredients on commercial farms and feed mills, other studies have failed to definitively link potentially contaminated swine feeds to transmission of ASFV [9], PEDV [13,14,15,16], and PRRSV [17]. Therefore, there is limited evidence from case studies showing clear linkages between feed contaminated with swine viruses and disease outbreaks on farms. Unfortunately, there is no global surveillance, monitoring, and testing program to determine the prevalence, frequency, concentrations, viability, and infectivity of these viruses throughout feed ingredient supply chains. As a result, there is high uncertainty about the relative risk of virus transmission through feed compared with other fomites and routes.

Because of the high uncertainty, mathematical models have been developed to evaluate the likelihood of virus transmission through feed. Galvis et al. [17] evaluated the relative likelihood of PRRSV transmission from nine transmission pathways and showed minimal association of feeding animal by-products on PRRSV outbreaks on farms. Schambow et al. [18] developed a quantitative risk assessment model to estimate the probability that one or more shipping containers with ASFV-contaminated soybean meal or corn would be imported to the U.S. annually. Although there was high uncertainty among many assumptions in this model, one container of corn contaminated with ASFV was estimated to be imported once every 50 years, but for soybean meal, the likelihood ranged from once every 21 to 1563 years. Other risk assessments for ASFV contamination of feed ingredients have been quantitative but have provided no uncertainty estimates [19] or have been qualitative without considerations for potential differences among ingredients [20,21]. Jones et al. [21] conducted a qualitative risk assessment for feed as a vehicle for transmission of prions, parasites, as well as several bacterial and viral pathogens, but provided no uncertainty estimates, and reported negligible overall risk for all pathogens except *Salmonella enterica*, PEDV, and ASFV. However, accurate interpretation of results from this study is difficult because no uncertainty estimates nor detailed assumptions used to make these determinations were provided.

Several laboratory-based inoculation studies have shown that most swine viruses of concern can survive in some feed ingredients for several weeks or months [12,22,23,24,25,26,27]. However, risk of virus transmission is based on the presence of a hazard (virus) and exposure to the host (pig). Before feed can be a source of infection, it first must be contaminated with a viral pathogen; then, the virus must survive the time and temperature conditions of drying, processing, and storage; the virus must survive during transport and subsequent storage at a feed mill; a feed ingredient contaminated with viable virus must be added to complete feed at a relatively high inclusion rate to provide virus concentrations greater than the minimum infectious dose in the final diet; and viruses must be in a viable form that can lead to infection when consumed by the pig. Therefore, there are many virus survival conditions that must be maintained from the time of an initial contamination event of a feed ingredient until adequate quantities of viable virus is consumed by pigs on a farm to ultimately cause infection and then disease. However, it is also important to recognize that viruses can cause asymptomatic or covert infections in pigs, making them carriers, and therefore there is an important distinction between infection and disease.

Because of the global trade of feed ingredients and subsequent movement of imported ingredients to feed mills and swine farms, feed biosecurity programs to minimize the risk of virus contamination and transmission have emerged as a relatively new component of feed safety and biosecurity protocols. Approaches to risk management of ASFV transmission through imported feed ingredients vary among countries [28,29]. In the U.S., all importers of food and feed ingredients are required by the U.S. Food and Drug Administration (FDA) to have a Foreign Supplier Verification Plan in place, but complete compliance with this legal requirement is questionable. In addition, all feed ingredient and complete feed manufacturers in the U.S. must have a Hazard Analysis and Risk-based Preventive Controls (HARPC) Feed Safety Plan in place, which includes identification of potential hazards and written plans to prevent contamination. Before 2022, the U.S. FDA did not consider viruses as a reasonably foreseeable hazard in animal feed, unlike pathogenic bacteria, fungi, and parasites. However, the FDA HARPC for Food for Animals Guidance for Industry (#245) now includes viruses as reasonably foreseeable hazards in feed supply chains, which requires developing a preventive control plan for preventing and monitoring virus contamination in feed. Interestingly, the U.S. FDA has classified viruses as the least heat resistant among potential types of microbial contaminants, which is not a scientifically valid assumption for many swine viruses, especially for ASFV. Based on the recent guidance, it is unclear if a preventive control plan for ASFV or any other swine viruses is required for feed manufacturers. It appears that the need to develop and implement a preventive control plan will be determined by a company’s perspective of risk as probable or possible to occur in time if not corrected. Regardless, there is no system nor standardized analytical procedures at commercial laboratories to facilitate and accurately measure viable virus contamination in feed ingredients and complete feeds to comply with the monitoring and corrective action requirements of HARPC. As a result, there are many knowledge gaps that need to be addressed to improve our ability to prevent and monitor virus contamination in swine feed, including:Identifying conditions during production, processing, transportation, and storage that can lead to virus contamination of feed ingredients;Determining the likelihood of swine virus contamination in feed;Understanding the chemical and physical characteristics of feed ingredients that allow various types of viruses to survive;Understanding the unique characteristics of various types of viruses that enable their survival and make them vulnerable to inactivation and loss of infectivity;Developing and validating highly sensitive and specific assays that accurately quantify viable and infectious virus particles for various viruses in different types of feed ingredients;Identifying time and temperature conditions that effectively inactivate viruses without degrading the nutritional value of ingredients;Identifying chemical mitigants that effectively inactivate viruses without degrading the nutritional value or safety of ingredients;Determining effective practices for decontaminating feed mills;Determining minimum concentrations of viruses and feeding conditions that prevent disease when pigs consume contaminated feed.

Because of our inability to accurately and routinely determine the presence, concentration, viability, and infectivity of virus contamination in feed ingredient supply chains, this high uncertainty of potential virus contamination requires development and implementation of biosecurity protocols that focus on specific swine viruses of concern. However, until more strict biosecurity regulations and programs are implemented, effective mitigation strategies are also needed to reduce viral load and viability in feed ingredients suspected of being contaminated. 

## 2. Identifying Production, Processing, Storage, and Transportation Conditions That May Cause Virus Contamination in Feed Ingredients and Complete Feed

The biosecurity of swine farms is an essential component for preventing introduction and controlling pathogens that cause foreign and endemic disease, and for maintaining high health to optimize productivity. International biosecurity guidelines have been developed that involve minimizing the exposure of animals to external hazards that are potential routes of pathogen transmission [30]. Major risk factors for pathogen transmission include airborne transmission; animal manure and soiled bedding; direct animal-to-animal contact; semen; human contact including dirty boots, clothing, and hands; zoonotic pathogens that are communicable between animals and humans; vehicles and other fomites; vectors including rodents, birds, insects, and feral animals; mortality disposal methods and equipment; and feed [31]. Unfortunately, biosecurity protocols of feed supply chains have generally not been included in overall biosecurity plans for swine farms, even though feed is a major external input and has some inherent risk for pathogen contamination and transmission to pig farms. 

Feed manufacturing facilities are a collection, storage, proportioning, mixing, and processing point for various types of feed ingredients sourced from many geographic regions before finished complete feeds are delivered to multiple farms. Therefore, the biosecurity of feed mills must be a major part of the overall biosecurity program for swine farms to prevent pathogen introduction. Cochrane et al. [32] described key components of developing feed mill biosecurity plans, which include: hazard analysis, which involves identifying and evaluating potential hazards in process steps used in the production of feed ingredients; hazard mitigation, which includes steps to prevent hazard entry during receiving, entry due to people, as well as cross-contamination during production, load-out, and delivery; use of thermal treatment such as pelleting; and the use of approved chemical treatments such as formaldehyde, essential oils, and medium-chain fatty acids (MCFAs). 

Feed manufacturers are responsible for the biosecurity of the feed supply chain, which begins with sourcing, receiving, and processing feed ingredients used to manufacture complete feeds until the delivery of finished feed to swine farms [33]. The American Feed Industry Association (AFIA) developed a working definition of a biosecure feed facility, which is a facility that has adopted procedures to reduce the risk of pathogens being transmitted into or contaminating final animal feed products [33]. In addition to AFIA guidelines, detailed standard operating procedures for good agricultural practices, good manufacturing practices, sanitary transport, and good warehousing practices for feed ingredients need to be developed and widely implemented. Biosecurity procedures may vary depending on the type of animal feed product produced, the disease status of the country or region where the feed manufacturing facility is located, and the source of ingredients used at the facility. A biosecurity plan for a feed manufacturing facility should include:Mechanisms for evaluating the quality, safety, and biosecurity procedures used by suppliers in the production of ingredients, including auditing and verification that protocols are followed;Facility design and maintenance protocols that prevent or reduce the introduction of pathogens;Routine housekeeping procedures that adequately prevent or reduce the introduction of pathogens;Standard operating procedures (SOPs) and surveillance programs for biosecurity that include ingredient sourcing, receiving, and storage;Biosecurity and personal hygiene protocols for visitors, employees, and drivers to control access to the facility;Manufacturing practices that are effective for maintaining the biosecurity protocols of the facility;Biosecure transportation of finished feed using sealed containers and disinfection practices.

Sanitary transport is commonly overlooked in biosecurity protocols. However, trucks, driver shoes, bags, and totes have been identified as the primary route of pathogen transmission associated with feed in several studies [9,13,14,17]. Cleaning, disinfecting, and heating trucks and trailers used for transporting pigs and feed between loads must be essential activities in feed supply chain biosecurity protocols. Disinfectants used to inactivate bacteria may not be effective for inactivating environmentally resilient viruses such as ASFV. As a result, several disinfectants have been evaluated for their efficacy in inactivating ASFV on various environmental surfaces [34]. Minimum heating time and temperature of transport vehicles has been evaluated by van Kessel et al. [35], where complete inactivation of several viruses (PEDV, PRRSV, swine influenza virus, transmissible gastroenteritis virus, and porcine rotavirus) and bacteria occurred when heated at 75 °C for a minimum of 15 min. However, the presence of fecal matter required longer heating times to achieve complete pathogen inactivation. Sanitary transport requirements for vehicles and transport equipment, transportation operations, training, and records for animal feed ingredients have been established in the U.S. Food Safety Modernization Act (https://www.fda.gov/food/food-safety-modernization-act-fsma/fsma-final-rule-sanitary-transportation-human-and-animal-food) (accessed 26 March 2023). Biosecurity protocols for sanitary transport of imported ingredients should include HARPC plans to reduce the risk of adulteration or cross-contamination of viruses. Key components of sanitary transport protocols include:Documentation verifying that the manufacturing and storage facilities in the country of origin have been decontaminated;One-way driveways for dirty vehicles and containers should be used to separate potentially contaminated vehicles and containers from those that are empty, clean, and disinfected using approved and effective disinfectants;Washing and disinfection facilities should be provided, and their use required for all trucks and equipment used for feed transport;After disinfection, transport vessels should be loaded and sealed at the manufacturing facility before transport to the destination;After ingredients are loaded and sealed, trucks should enter the delivery destination through a “clean” driveway;After unloading, transport time and temperatures conditions should be recorded and considered when estimating required holding times during storage at the destination;Upon arrival at the destination, only trucks that are empty, clean, and disinfected should be used to transport bulk ingredients for quarantine in a heated temporary warehouse;For bagged ingredients, new or properly cleaned and disinfected pallets should be used;Documentation of storage conditions and holding times for each lot of each feed ingredient should be provided to end users.

Upon completion of a comprehensive feed mill biosecurity plan, all new procedures must be correctly implemented to reduce the likelihood of pathogen introduction. Therefore, employee training must be provided so that they can demonstrate an understanding of each risk factor being controlled and that they are capable of following procedures and protocols to minimize each risk. Unannounced internal audits are useful to ensure compliance with existing procedures and to identify aspects of protocols where more employee training is needed. Biosecurity plans should be re-evaluated at least once annually or when a new feed ingredient source is acquired at the feed manufacturing facility. External third-party audits, review, and consultation from qualified feed mill biosecurity experts are also useful practices for identifying potential hazards that may have been overlooked when initial protocols were developed.

## 3. Challenges of Measuring Virus Inactivation

Accurate and repeatable analytical assays for routine evaluation and monitoring of common bacterial pathogens including *Salmonella*, *Escherichia coli*, and *Clostridium perfringens* contamination in feed samples have been routinely used for many decades [36]. Virus-specific and sensitive assays for routine use in measuring viability and infectivity of viruses in feed have been adopted from various cell-culture-based in vitro assays such as TCID_50_ [37] and hemadsorption tests (HAD_50_) [38] and in vivo animal infection bioassay procedures [25]. Unfortunately, there are no standardized analytical procedures at commercial laboratories to facilitate the monitoring and accurate measurement of viable virus contamination in feed ingredients and complete feeds to comply with the monitoring and corrective action requirements of HARPC. 

Accurate determination of virus concentrations in feed ingredients begins with collecting representative samples. However, viruses in contaminated feed may not be uniformly distributed and may be present in low concentrations. No sampling methods have been validated for use in collecting representative feed samples for virus analysis. Jones et al. [39] inoculated soybean meal samples with 10^3^ TCID_50_/g or 10^5^ TCID_50_/g PEDV and collected samples using individual probes or composite sampling. These researchers reported that composite samples were more sensitive to virus RNA than probes in bulk soybean meal and suggested a minimum of 10 subsamples be collected for creating a composite sample for analysis. Elijah et al. [40] evaluated the use of a “double X pattern” sampling procedure to collect subsamples for determining ASFV concentrations in bulk ingredients using the procedure described by Jones et al. [39] and suggested that collection of 10 subsamples was necessary to obtain accurate results.

There are also numerous challenges with using existing analytical assays for the accurate detection of viable virus particles in feed ingredients that can lead to infection, especially for ASFV [41]. Measuring virus viability represents a higher standard of quantification than relying on the detection of nucleic acids via polymerase chain reaction (PCR) analysis. If virus viability can be eliminated, then infectivity will be prevented. Infectivity assays are not robust and are subject to creating misleading data. Virus particles need to interact with the host cells to complete the biological process of replication into new virus progeny. The virus–host cell interaction can yield a diverse set of changes in cell physiology with the aim of producing viral progeny. Therefore, the term “infection” is generally used to refer to the production of new virus particles. Because the most common methods of virus quantification rely on observation of cell death, infectivity is measured by the observed cell death. Viruses can enter a cell, replicate, and make changes to cell physiology, yielding progeny but without observable cell death; in this case, the virus can be classified as viable but not infectious [42].

Targeted virus diagnostic methods must have high specificity (accurately identify negative results) and high sensitivity (accurately identify positive results) to accurately quantify the amount of viable virus particles capable of causing infection if ingested by pigs [41]. Results from most analytical measures are often erroneously extrapolated to infer virus infectivity, which causes a false assessment of virus inactivation and capability of causing disease. Although conventional qPCR assays are commonly used to determine the amount of virus nucleic acid (RNA or DNA) copies per unit of volume or weight in a feed sample, their use is associated with challenges that can lead to misinterpreting the results. Research studies often provide Ct values (cycle time), which is the number of PCR cycles needed to detect viral RNA or DNA. A Ct value is the number of amplification cycles needed to reach a fixed background level of fluorescence at which the determined results change from negative (non-detectable) to positive (detectable). Common methods for handling qPCR non-detectable nucleic acids lead to biased inferences [43]. The total number of cycles required to exceed a pre-determined threshold for a positive result can range from 15 to 40 cycles but is specific to the test platform being used. Furthermore, various tests count the number of cycles and calculate Ct values differently. In general, Ct values less than 40 are acceptable for determining the presence or absence of viral nucleic acids, but for quantification, only Ct values less than 35 should be considered reliable. Although there is a relationship between Ct values and the amount of virus in a sample, they are not equivalent because many feed sample collection and analytical variables affect Ct values. Variables involving feed sample collection include: (1) obtainment of a representative sample, (2) time of collection after contamination, (3) type of feed ingredient matrix, (4) concentration of nucleic acids in the sample, and (5) storage and transport conditions of sample prior to testing. Variables that affect the analysis include: (1) nucleic acid extraction efficiency, (2) amount of viral nucleic acids in the samples, (3) design of primer/probe sequences, (4) efficiency of chemistry in the assay, and (5) method for determining Ct value. In addition, qPCR assays are qualitative and do not distinguish between free DNA from damaged viruses and DNA from intact viable virus particles, do not assess virus infectivity, and may not always correlate with viral concentrations. Because Ct values are assay specific, comparison of Ct values across assays can also lead to misleading interpretations. 

For some viruses, PCR methodologies have been modified to create viability PCR assays that distinguish between viable and free DNA in feed samples [44]. However, depending on the virus, viability does not always equate to infectivity [45]. Virus isolation assays are also frequently used in research studies [23] to assess virus infectivity in cell culture, but they also do not infer virus infectivity if ingested by pigs [46]. The hemadsorption test (HAD_50_/mL) is another common assay used to calculate virus concentrations by incubating a red blood cell suspension with an infected cell culture to measure 50% of the replicates, showing the amount of virus hemadsorption per milliliter of blood [47]. However, this method has been shown to be not adequately sensitive for definitive ASFV diagnostics [48]. The most common assay for assessing virus infectivity is the median tissue culture infectious dose (TCID_50_) per gram of sample, which measures the amount of virus capable of infecting 50% of cell culture replicates, but it does not directly infer infectivity in pigs. More recently, surrogate assays have been developed for some swine viruses in research applications because using the actual virus of interest, such as ASFV [45], CSFV, or FMDV [23], is restricted to high-biosecurity facilities. Although surrogate assays show great promise as cost-effective and rapid alternatives to the more traditional assays, they have yet to be widely implemented and thus tested, to ensure that their sensitivity and applicability for assessing the infectivity of the specific viruses they are meant to simulate is acceptable. Pig bioassays have been used in some studies [23] to confirm that positive PCR results in feed samples are capable of causing infection, but bioassays are limited by scale, require the use of high-biosecurity animal facilities, are expensive and time consuming to conduct, and may not provide reliable, consistent results [41]. 

In addition to understanding the specific state of viruses being measured by various assays, accurate interpretation of results from virus inactivation studies is also essential. The term “complete inactivation” of viruses should be avoided because it infers zero risk of infection in pigs consuming feed that was originally contaminated, which is not possible [41]. Although studies have been conducted to estimate the minimum infectious doses of various swine viruses to cause disease, there are data inconsistencies regarding the number of virus particles necessary to cause infection because of inaccuracies in the current diagnostic assays used. These inaccuracies, along with the need to consider the number of animals in a population that may be exposed to a virus, are major factors that determine the likelihood of a disease outbreak. Inactivation data in feed ingredients are often described as a 99.9% reduction of viruses, which corresponds to a 3-log or 10^3^ reduction from the initial virus concentration and should not be interpreted as 0.1% virus particles remaining in the sample. Therefore, depending on the initial virus concentration, a 99.9% or 3-log reduction in virus from thermal or chemical mitigation treatments may exceed the minimum infectious dose for a contaminated feed and potentially result in an infection. In addition, virus inactivation kinetics curves usually fit non-linear patterns with shoulders or tails and do not follow first-order kinetics (log-linear). Half-life (time to reduce 50% of the initial virus concentration) data for virus inactivation reported in some studies does not provide an accurate estimate of virus inactivation in a feed ingredient because it is based on the assumption of a linear relationship between the logarithmic decrease in virus concentration and time of mitigant exposure. However, using inactivation data to calculate D-values (amount of time needed to reduce initial virus concentration by 1 log or 90%) and z-values (change in temperature needed to achieve a 1 log or 10 times reduction of the D-value) are useful to make comparisons of virus inactivation kinetics across studies and predict outcomes not previously determined experimentally. Unfortunately, most studies have not provided adequate data suitable for calculating D- and z-values to more accurately determine virus inactivation kinetics in feed ingredients [41]. In summary, substantial limitations exist among various virus detection assays, which makes it difficult to compare results across studies and accurately assess virus viability and infectivity in feed ingredients and complete feeds.

## 4. Virus Survival in Feed Ingredients during Transport

Several studies have evaluated virus survival in feed ingredients under various types of transport conditions. The initial study was conducted by Dee et al. [23] and included six feed ingredients experimentally inoculated with one of 11 swine viruses or surrogates (including ASFV, CSFV, PEDV, SVV-A, and PRRSV) and stored under temperature and relative humidity conditions to simulate 30-day trans-Atlantic or 37-day trans-Pacific shipping conditions. Infectious viruses were recovered for SVV-A, ASFV, PEDV, and PRRSV in conventional soybean meal; ASFV and PEDV in organic soybean meal; SVV-A and PRRSV in DDGS; SVV-A and PEDV in L-Lysine HCl; and SVV-A, ASFV, and PEDV in choline chloride. These results indicate that viruses survive in feed under simulated trans-oceanic transport conditions, but survival varies among viruses and feed ingredient matrices, with conventional soybean meal appearing to have chemical and physical properties that support the survival of the most viruses. In a follow-up study, Stoian et al. [24] used the same ingredients and 30-day trans-Atlantic shipping conditions as those used by Dee et al. [23] to calculate the half-life of ASFV, which ranged from 14.2 days in complete feed to 9.6 days for conventional soybean meal, with an average overall half-life for all ingredients of 12.2 days. Another study evaluated 12 feed ingredients that were inoculated with CSFV and pseudorabies virus (PRV) and exposed to simulated environmental conditions of a 37-day trans-Pacific shipment model [49]. Infectious CSFV was detected by pig bioassays in conventional soybean meal, infectious PRV was found in L-lysine HCl and choline chloride, and cell culture titers of PRV were found in conventional and organic soybean meal and vitamin D on day 37. 

Simulations of long-distance truck transport of virus-inoculated feed ingredients and complete feed in the U.S. have also been evaluated. Dee et al. [50] conducted a demonstration study to determine if viable and infectious PRRSV, PEDV, and SVV-A would survive a 21-day commercial truck transport for more than 9000 km across 14 states in the U.S. when a mixture of these viruses was inoculated in organic and conventional soybean meal, L-lysine HCl, choline chloride, and vitamin A. All viruses were detected as infectious in soybean meal, while infectious SVV-A was found in L-Lysine HCl and vitamin A. Using the same experimental design and 23-day commercial U.S. truck transport model, Dee et al. [25] showed that inoculation of conventional soybean meal, organic soybean meal, and conventional feed with PRRSV, PEDV, and SVV-A resulted in all viruses remaining infective at the end of the transport period based on pig bioassays. In addition, a surrogate virus (Emiliania huxleyi virus or EhV) for ASFV was used to inoculate conventional and organic soybean meal and swine complete feed to simulate an ASFV contamination event [51] using the same experimental design as described by Dee et al. [50]. Results showed that viable EhV was detected in all matrices at the end of the transport period, and no degradation of viability occurred. In summary, results from these simulated and real-world studies show that the time, temperature, and relative humidity conditions of trans-oceanic and trans-United States transport do not reduce viability and infectivity of PEDV, PRRSV, SVV-A, and a surrogate for ASFV.

## 5. Virus Inactivation of Various Feed Ingredients during Extended Storage

Several studies have evaluated the use of extended storage time as a mitigation strategy to determine the rate and extent of inactivation of various swine viruses in different feed ingredients during various time and temperature exposures, and the results are summarized in Table 1. Studies evaluating virus survival during transport [23,24,50,51] were excluded from this summary because temperature and relative humidity conditions varied over time. Similarly, because of the extremely low and variable storage temperatures (−18 °C for days 1–7, −13 °C for days 8–14, and −9 °C for days 15–30) used in a storage study evaluating PEDV survival in 18 feed ingredients during a 30-day storage period [22], determination of accurate virus inactivation rates for various feed ingredients at different temperatures is not possible or meaningful.

Although extended storage times have been shown to be a simple and effective way of reducing virus concentrations in feed ingredients and complete feeds if they are contaminated, there are several unintended consequences that must also be considered. Storage facilities for feed ingredients represent a significant cost for feed handling and manufacturing facilities. As a result, frequent inventory turnover is required to minimize the cost of ingredient procurement and feed manufacturing. Furthermore, depending on the time, temperature, and relative humidity conditions of various types of feed storage and the physical and chemical characteristics (e.g., moisture content and water activity) of ingredients, significant feed safety concerns and loss in the nutritional value of ingredients can occur. High temperature, humidity, and moisture conditions cause bacteria growth and mold and mycotoxin production during storage that can be detrimental to animal health and performance. These same conditions can also lead to significant losses of nutritional quality and value through reduced protein and amino acid digestibility, production of secondary lipid oxidation products, and loss of vitamin potency. Therefore, maintaining a balance between minimal storage time to reduce cost and preserve nutritional value and allowing adequate time for significant virus inactivation must be considered when using this mitigation strategy to inactivate swine viruses in contaminated feed ingredients.

Dee et al. [52] indicated that a 30-day storage period at ambient temperature has become a standard recommendation for inactivating all swine viruses in all feed ingredient matrices for voluntary [53] and government programs in Canada [29] for imported feed ingredients. However, this recommendation is based on limited data, many unfounded assumptions, different analytical techniques and measures, and extrapolations from studies that did not assess virus infectivity. Therefore, there are many data gaps, inconsistencies among findings, and concerns for adopting this recommendation. Other government programs in Australia (https://www.agriculture.gov.au/import/goods/plant-products/stockfeed-supplements) (accessed 26 March 2023) and the European Union (https://fefac.eu/wp-content/uploads/2020/07/recommendation_biosecurity_v10_final-1.pdf) (accessed on 26 March 2023) also provide guidelines for assessing the risk of ASFV in imported feed ingredients but do not specifically require a standard 30-day storage period for suspect ingredients.

**Table 1 animals-13-02375-t001:** Summary of published studies evaluating effects of storage time and temperature for inactivating swine viruses in various feed ingredients.

Virus	Feed Ingredients	Temperature-Time	Assays Used	Reference
African swine fever	SDPP *	4 °C or 21 °C for up to 35 days	Hemadsorption tests, real-time PCR, cell culture for virus isolation	[54]
	Soybean meal, ground corn cobs, complete feed	4 °C, 20 °C, or 35 °C for up to 365 days	TCID_50_/mL, cell culture for virus isolation, pig bioassay	[27]
Classical swine fever	No studies have been conducted	No data	No data	-
Foot and mouth disease	DDGS **, soybean meal, complete feed	4 °C or 20 °C for up to 37 days	Half-life	[55]
Porcine epidemic diarrhea virus	SDPP	4 °C, 12 °C, or 22 °C for up to 21 days	TCID_50_/mL, cell culture for virus isolation	[56]
	Conventional soybean meal, organic soybean meal, choline chloride, L-lysine HCl, vitamin A	Indoor: −20 °C for 30 days; outdoor: −4 °C to −14.7 °C (avg. −8.8 °C) for 30 days	PCR, pig bioassay	[57]
Porcine epidemic diarrhea virus, porcine delta corona virus, transmissible gastroenteritis virus	Corn, low-oil DDGS, medium-oil DDGS, high-oil DDGS, soybean meal, SDPP, blood meal, meat meal, meat and bone meal, vitamin-trace mineral premix, complete feed	25 °C for up to 56 days	TCID_50_/mL, cell culture for virus isolation, delta values	[58]
Porcine reproductive and respiratory syndrome virus	Soybean meal	10 °C, 15.5 °C, or 23.9 °C for up to 30 days	PCR of oral fluid, pig bioassay	[52]
	Conventional soybean meal, organic soybean meal, choline chloride, L-lysine HCl, vitamin A	Indoor: −20 °C for 30 days; outdoor: −4 °C to −14.7 °C (avg. −8.8 °C) for 30 days	PCR, Pig bioassay	[57]
Seneca Valley A virus	DDGS, Soybean meal, Vitamin D, L-lysine HCl	4 °C, 15 °C, or 30 °C for up to 92 days	TCID_50_/mL, half-life, reverse transcriptase rt-PCR, pig bioassay	[26]
	Soybean meal	10 °C, 15.5 °C, or 23.9 °C for up to 30 days	PCR of oral fluid, pig bioassay	[52]
	Conventional soybean meal, organic soybean meal, choline chloride, L-lysine HCl, vitamin A	Indoor: −20 °C for 30 days; outdoor: −4 °C to −14.7 °C (avg. −8.8 °C) for 30 days	PCR, pig bioassay	[57]

* SDPP = spray-dried porcine plasma; ** DDGS = corn distillers dried grains with solubles.

The number of studies assessing the survival of several of these important swine viruses in numerous common feed ingredients used in swine diets under different time and temperature conditions are limited. No studies have been conducted to evaluate CSFV survival during storage under controlled time and temperature conditions. Estimates for FMDV, porcine delta corona virus (PDCoV), transmissible gastroenteritis virus (TGEV), and PRRSV survival have each been determined in only one study, and the ingredients evaluated were limited to only soybean meal for PRRSV and soybean meal, DDGS, and complete feed for FMDV. Two studies have evaluated ASFV survival in four ingredients and an additional two studies have evaluated SVV-A survival in four ingredients. Although these are critically important studies to provide initial guidance on recommended storage times for virus inactivation of ingredients suspected of being contaminated, the data available from these studies is insufficient for supporting the 30-day extended storage time recommendation proposed by Dee et al. [52]. All of the studies evaluating multiple feed ingredients have shown distinct differences in virus survival and inactivation among ingredients at various times and temperatures, indicating that different recommendations are needed for different viruses and different types of ingredients. A standard definition of acceptable virus inactivation needs to be defined based on the capability of residual virus to cause infection because different analytical methods vary in sensitivity and specificity, leading to inconsistent results and interpretation within and among studies. 

To better understand the significance of the inconsistencies reported for extended storage time studies, consider the different methods, ingredients, and interpretation of results between the two studies evaluating ASFV in Table 1. Fischer et al. [54] indicated that storing spray-dried porcine plasma (SDPP) at 21 °C for 14 days results in complete inactivation (>5.7 log reduction) based on hemadsorption tests and cell culture assays. In contrast, Niederwerder et al. [27] recommended that complete feed and feed ingredients be stored for >112 days at 4 °C, >21 days at 20 °C, and <7 days at 35 °C to reduce the risk of ASFV infection in pigs. This recommendation was based on detecting infectious ASFV until 112 days at 4 °C in soybean meal despite <60 days at 4 °C being needed for complete feed and <7 days at 4 °C for corn cobs. These dissimilar results for different feed matrices using different assay and time and temperature conditions make it difficult to have confidence in the effectiveness of extended storage time protocols for adequately inactivating ASFV and certainly do not support the 30-day standard proposed by Dee et al. [52]. 

Feeding frequency of pigs used in bioassays and virus strain were additional factors beyond time, temperature, and assay methods that affected results for FMDV survival in soybean meal, DDGS, and complete feed during a 37-day storage period [55]. The last timepoint of detectable FMDV in this study was at 37 days for both FMDV strains evaluated in soybean meal regardless of temperature but ranged from 3 to 14 days in complete feed at 20 °C depending on the virus strain, and DDGS was determined to be highly toxic in cell cultures, resulting in an estimated half-life of one hour. 

Extrapolating minimum storage times for virus inactivation is difficult if storage temperatures are below those commonly found in warm climates. For example, one extended storage study was conducted under conditions of extremely low and variable storage temperatures (−18 °C for days 1 to 7, −13 °C for days 8 to 14, and −9 °C for days 15 to 30) to evaluate PEDV survival in 18 feed ingredients during a 30-day storage period [22]. Because temperature has a major effect on virus inactivation rates for various feed ingredients, it is not appropriate to apply these results to feed mill environmental conditions during warm and hot months of the year. However, under these relatively cold conditions, viable PEDV was detected by virus isolation or swine bioassay up to 30 days in soybean meal, DDGS, meat and bone meal, spray dried red blood cells, L-lysine HCl, DL methionine, choice white grease, choline chloride, and complete feed. Viable virus was also found in ground limestone up to 7 days, and in L-threonine up to 14 days after inoculation. In contrast, viable PEDV was detected in soybean meal and complete feed for up to 180 and 45 days, respectively, during an extended storage evaluation period.

Extrapolating virus survival and inactivation data for a specific virus of interest from data derived from other virus types has been done, but it is not advised because of the uniqueness of each virus and the complexities of their behavior in various feed matrices. For example, PDEV, PDCoV, and TGEV are in the *Coronaviridae* family, of which PEDV and TGEV are classified in the Alphacorona virus genus, and PDCoV is in the Deltacorona virus genus [59]. These coronaviruses are enveloped, single-stranded-RNA viruses with a size of about 30 kb and are structurally similar [59]. However, despite these similarities, Trudeau et al. [58] showed that their inactivation kinetics in the same ingredients and under the same time and temperature conditions are quite different. Results from this study showed that D values for SDPP varied from 1.14 to 3.25 to 19.18 days for PEDV, PDCoV, and TGEV, respectively. Soybean meal had the lowest D value for PEDV (7.50), compared with 42.04 and 41.94 for PDCoV and TGEV, respectively, while corn also had greater D values for PDCoV (25.60) and TGEV (11.78) compared with PEDV (2.25) and all other ingredients except soybean meal. Furthermore, none of the three coronaviruses evaluated in this study were completely inactivated after a 56-day incubation period. Estimated inactivation of viruses in soybean meal ranged from 23 days for PEDV to 126 days for PDCoV and TGEV at the same temperature (25 °C). Therefore, extrapolation of inactivation times from data evaluating other viruses is not appropriate. 

One cannot extrapolate virus inactivation results for individual feed ingredients or complete feed from studies that used non-feed matrices or cell cultures. Knight et al. [60] summarized results from studies that determined thermal inactivation of ASFV, CSFV, and FMDV in non-feed matrices using plaque assays, hemadsorption assays, and animal models, and reported very different inactivation conditions than observed for feed ingredients. For example, multiple studies evaluating FMDV inactivation in the Knight review [60] showed relatively high temperatures of 54 °C to 110 °C being required to inactivate virus depending on the experimental matrix and method of evaluation used. Furthermore, these temperatures are prohibitive for inactivating FMDV in feed ingredients if nutritional value is to be preserved. 

Dee et al. [52] evaluated three storage temperatures (10 °C, 15.5 °C, and 23.9 °C) during a 30-day storage period on SVV-A infectivity of soybean meal using natural pig feeding behavior with the “goal of providing data for the development of industry standards for the management of high-risk ingredients”. This goal cannot be achieved by limiting the storage period to 30 days and focusing only on soybean meal. Using their “hot spot model”, 10 mL ice cubes containing 10^5^ TCID_50_ of SVV-A and the same concentration of PRRSV were added to feed ingredients. Results showed that only the storage temperature of 23.9 °C was effective in achieving no infectivity after 30 days. In contrast, Caserta et al. [26] inoculated feed ingredient samples with 10^5^ TCID_50_ of SVV-A and showed that virus inactivation in L-lysine HCl and vitamin D occurred within 1 day at 15 °C and 30 °C compared with soybean meal and DDGS, which survived until 35 days at 15 °C and until 21 and 14 days, respectively, at 30 °C. Although Caserta et al. [22] did not confirm the infectivity of soybean meal and DDGS in pig bioassays, their data clearly showed that a 30-day extended storage period to inactivate SVV-A in L-lysine HCL and vitamin D is excessive and may result in loss in vitamin D potency.

Although soybean meal appears to be most protective for swine virus survival, it has also been the most studied of all ingredients. Other grains, grain by-products, and oilseed meals must also be evaluated for their ability to protect various swine viruses if they become contaminated because they are added to diets at much greater inclusion rates than vitamins and synthetic amino acids, and subsequently could contribute a greater virus dose for pigs to consume. However, results from several studies evaluating the survival of several viruses in soybean meal have shown that more than 30 days storage, even at relatively low temperatures, is necessary to achieve significant virus inactivation. The likelihood of a virus-contaminated ingredient causing infection is not only dependent on the initial virus concentration used to inoculate feed ingredients, and the subsequent extent of virus inactivation and loss of infectivity that may occur during storage, but also on the frequency of consumption of contaminated feed (i.e., once versus multiple feeding events) [55,61] that exceed the minimum infective doses of each virus of interest.

Structural and chemical characteristics of various feed ingredients likely play a role in the differences in survival of various viruses among different feed ingredients, but these factors have not been studied. The moisture content of feed ingredients and complete feeds is relatively low (<12%), but Trudeau et al. [58] reported that increasing moisture content of ingredients was moderately correlated with increased survival of PDCoV (r = 0.48) and TGEV (r = 0.41). However, water activity may be a better indicator of virus survival in feed ingredients than moisture content because it has been shown to more accurately predict the likelihood of microbial growth in foods [62]. Water activity is the amount of unbound or available water in foods on a scale between 0 and 1, and it has been shown to be a primary factor attributed to the thermal resistance of bacterial pathogens [63], where values below 0.60 are generally considered adequate for preventing bacterial and mold growth in foods [64]. Although the water activity has not been extensively evaluated in feed ingredients, Hemmingsen et al. [65] reported that coarse-milled soybean meal had greater water activity than finely ground soybean meal, and coarse or finely ground barley, rapeseed cake, and corn. These results suggest that particle size affects the water activity of ingredients. In addition, oil content may be another chemical factor that affects virus survival in soybean meal. Studies comparing conventional and organic soybean meal have consistently shown greater virus survival in organic soybean meal (6 to 7% oil) compared to conventional soybean meal (1 to 2% oil). More research is needed to understand the relative effects of various physical and chemical characteristics of ingredients on the survival of various swine viruses, which may be useful for developing models that can predict inactivation rates under various time and temperature conditions of different feed matrices.

## 6. Minimum Infectious Doses

Estimates for minimum infectious doses (MID) of several swine viruses have been determined through consumption of inoculated feed or direct oral inoculation in pigs (Table 2). Virus strain appears to influence MID, as shown for CSFV [66] and FMDV [6], as well as age of pig for SVV-A [67]. Except for SVV-A, the MID is greater than 10^4^ for all other major swine viruses, which suggests that achieving a 3-log reduction in infectious virus concentration from an initially high theoretical contamination level of 10^6^ in complete feed would be below MID for all viruses except for SVV-A in market-age pigs. However, all of these MID estimates are based on observations from a relatively small number of pigs used in these studies and would likely decrease when estimated using a greater number of animals in a population of pigs.

The minimum infectious dose cannot be interpreted as a concentration in which a product, if contaminated, is considered safe, or the safe concentration from which no infection will occur. The MID in the experiments presented in Table 2 only represents the lowest concentration at which, under the conditions of the experiments, no animal presented signs of infection. However, animals may develop disease if a greater number of animals are exposed to contaminated feed than those used in these studies. In addition, results from a study with chicks showed that feeding diets with no detectable levels of *Salmonella* in feed can still cause an infection when fed [72]. This disparate phenomenon is possible because the likelihood of infection and risk of disease transmission increase with increased exposure to the pathogen and because of the greater frequency of feed consumption occurring in commercial swine herds compared to small experimental infections. After consuming feed contaminated with 10^7.0^ TCID_50_ of FMDV three times, all pigs (4 of 4) were infected after two days. Likewise, after consuming 10^7.2^ TCDI_50_ of FMDV only one time, all pigs (4 of 4) were infected [55]. These results demonstrate the importance of interpreting the infectious dose and frequency of consumption of contaminated feed data carefully. Another way to think about this concept is to simply consider the difference in risk of infection estimated at the individual level (small risk) versus risk at the group level (greater risk). The greater frequency of exposure of the group increases the likelihood of an adverse event. 

The minimum infectious dose is different from other estimates of food and feed safety such as the food safety objective (FSO) and performance objective (PO) [73]. These two methodologies allow the achievement of different goals in food safety and could be applied to the safety of feed ingredients and complete feed contaminated with swine viruses (Table 3). However, there are currently no published studies that calculate FSO or PO for virus inactivation in feeds for pigs, but minimum infectious dose and microbiological risk assessment data are needed to address this challenge.

## 7. Virus Inactivation from Thermal and Irradiation Processes in Feed Ingredients

In addition to extended storage times, thermal processing can be an effective method of inactivating bacteria, viruses, and parasites depending on the temperature and duration [7]. Heat is commonly used when drying grains and manufacturing various types of feed ingredients and complete feeds. Historically, rendered animal by-products have often been perceived to be at greater risk for contamination and transmission of various biological agents compared with grain and grain-based by-products. However, grain, oilseed meals, and grain by-products can also be contaminated with pathogens. In the United States, dry rendering is the most common process used in either batch or continuous systems, where heat (120 °C to 135 °C) produced by steam condensation is applied and uniformly distributed to ground carcass material for 45 min to 1.5 h under pressure (2.8 to 4.2 bar) [74]. These thermal processing conditions are effective for completely inactivating several viruses including ASFV, CSFV, and FMDV in meat product matrices [60]. 

Most studies conducted with feed ingredients and complete feed have evaluated thermal processing on PEDV inactivation. A summary of the effects of various time and temperature conditions used during various types of production processes of feed ingredients and complete feed is shown in Table 4 [75]. Temperatures greater than 130 °C were effective in reducing PEDV survival in various feed ingredient matrices [58,76], and the time and temperature used during the spray drying of plasma protein was shown to be effective in completely inactivating the virus [77]. Conditioning and pelleting temperatures greater than 54 °C were effective in reducing the quantity and infectivity of PEDV in swine feed [78]. Furthermore, application of an irradiation treatment of 50 and 86.25 kGy to feed resulted in a 3 and 5 log reduction, respectively, in PEDV concentration [76].

Several studies have shown that the use of ultraviolet irradiation is an effective additional biosecurity step to further inactivate several enveloped (i.e., PRRSV, PEDV, SVV-A, CSFV) and non-enveloped swine viruses during the spray drying process of liquid porcine plasma [79]. In general, results from these studies showed that enveloped viruses are more sensitive to ultraviolet C irradiation than non-enveloped viruses, but infectivity is reduced by at least 4 logs. Furthermore, although spray drying effectively inactivates at least 4 logs of ASFV and CSFV, the use of ultraviolet C irradiation within the spray drying process can provide additional inactivation of ASFV by more than a 4 log TCID_50_/mL reduction [80]. 

Additional studies have been conducted to evaluate the effectiveness of the spray drying process used for animal plasma [56,77,79,81] to inactivate ASFV, CSFV, PEDV, and PRRSV (Table 5), and a few other studies have evaluated lab-scale drying and heating of several grains, DDGS, soybean meal, and animal by-products [58,82,83]. Only one study has evaluated conditioning and pelleting time and temperatures on PEDV inactivation of complete feed [78]. In general, results from these studies show that various time and temperature thermal treatments are effective in at least partially reducing viral concentrations in all feed matrices, but the magnitude of reduction varies considerably among types of ingredients evaluated, thermal processes used, initial virus concentrations, thermal sensitivity or resilience of the virus, and method of detection. As a result, additional mitigation strategies, such as the use of chemical mitigants, are needed to achieve greater assurances of virus inactivation in potentially contaminated feed ingredients and complete feed fed to swine.

## 8. Virus Inactivation from Chemical Mitigants in Feed Ingredient and Complete Feed Matrices

Limited studies have been conducted to evaluate various types of feed additives for their effectiveness as chemical mitigants for inactivating ASFV [85,86], FMDV [55], PDCoV [87], PRRSV, and SVV-A [88] (Table 6). Most of the chemical mitigant studies have focused on the efficacy of inactivating PEDV in feed ingredients and complete feed [76,88,89,90,91,92,93,94,95,96,97]. Of the various chemical mitigants evaluated, a commercial aqueous formaldehyde and propionic acid (FMPA) product has been the most extensively studied and has been shown to be one of the most potent and effective viricidal products for a least partial inactivation of all swine viruses considered to date. However, although this FMPA product is approved for use in controlling *Salmonella* in poultry and swine feed, it is not approved for use in controlling swine viruses in the U.S. and numerous other countries. Various individual MCFAs and MCFA blends have also been extensively evaluated for their potential viral mitigation effects. Unlike FMPA, MCFAs such as C6:0 (caproic acid), C8:0 (caprylic acid), C10:0 (capric acid), and C:12:0 (lauric acid) are naturally found in triglycerides present in common fats and oils used in animal feeds, and their use is generally not restricted in commercial swine feeds. A few commercial products that have been evaluated as chemical mitigants in virus-contaminated swine feed contain certain short-chain or long-chain fatty acids, but their potential viricidal effects are questionable. Glycerol monolaurate has been shown to have more potent viricidal effects than MCFA for ASFV [86], and some proprietary monoglyceride products also have been shown to have potent viricidal effects for PEDV [94]. In addition, several commercial products include various types of organic acids and acidifiers, such as lactic acid, phosphoric acid, citric acid, fumaric acid, and benzoic acid, that appear to provide beneficial partial inactivation of FMDV [55], PDCoV [87], PRRSV and SVV-A [88], and PEDV [76,88,97]. Other components of some commercial mitigant products include essential oils, prebiotic fiber, and bacterial fermentation products [88], which may provide some viricidal benefits, but their efficacy relative to FMDV and MCFA needs to be evaluated. Interestingly, the addition of sucrose and sodium chloride has also been shown to be partially effective for PDCoV [87] and PEDV [76] inactivation in complete feed. In general, results from these studies have shown that most of the feed additives evaluated provide some benefit for reducing swine virus concentrations, which is often based on a reduction in nucleic acid concentrations from PCR analysis. Future studies should utilize viability PCR as a more definitive measure to determine the presence or absence of viable virus resulting from mitigation treatments. Studies are also needed to evaluate the effectiveness of various combinations of extended storage time and temperature, thermal and irradiation processing, and chemical mitigants on the inactivation of various swine viruses in different feed ingredients matrices.

Among all swine viruses evaluated, ASFV is the most difficult to inactivate because of its high thermal tolerance [45]. Fortunately, some chemical mitigants appear to be promising for providing partial ASFV inactivation, such as FMPA, MCFA, and GML, but it is unclear if their application reduces the risk of infectivity enough when fed to pigs. More research is needed to fully understand the effectiveness of these chemical mitigation strategies, but this has been difficult to accomplish without effective surrogate viruses for ASFV and more refined molecular diagnostic tools. Furthermore, more extensive scientific exploration is needed to develop appropriate molecular-based diagnostic methods to better understand the extent and type of degradation of ASFV in swine feed that is necessary to prevent infection of pigs.

## 9. Effectiveness of Virus Decontamination Strategies in Feed Mills

Pathogen contamination can occur on feed and non-feed contact surfaces in feed mills if contaminated feed ingredients are introduced, despite the use of well-designed and well-implemented feed mill biosecurity protocols [36,98]. Because of the interconnectedness of individual feed mills serving multiple farms in large geographic areas, an additional potential source of pathogen transmission can occur via fomites associated with feed manufacturing and delivery personnel, vehicles, and equipment. Greiner [14] collected daily environmental samples from 24 commercial feed mills that delivered feed to infected swine farms to evaluate the prevalence of PEDV and PDCoV contamination using a standard qPCR test. Although these data do not indicate whether viable virus was present, it was used as a proxy for presence. Results from this study showed that while no feed mills tested positive for PEDV, there was a low prevalence (<5%) of contamination that occurred on truck foot pedals and bulk ingredient pits, with a similar low prevalence of 3.4% truck foot pedals and 2.2% of office floors suspected of contamination for PDCoV. A more comprehensive evaluation of the potential transmission routes for PRRSV has been conducted by considering nine pathways that included pig movements, farm-to-farm proximity, different transportation vehicle networks (including feed), and use of animal by-products in feed [17]. Results from this study showed that vehicles transporting pigs to farms had the greatest contributions to PRRSV infections, while feed delivery to farms and the use of low dietary inclusion rates of animal fat and meat and bone meal had no significant contribution to PRRSV transmission. Gebhardt et al. [9] collected environmental samples from pig production, feed manufacturing, and feed distribution systems in ASFV-infected areas in Vietnam to evaluate ASFV contamination using qPCR analysis. Results from that study showed very low prevalence of ASFV-positive samples from feed delivery vehicles (0.69%), feed and non-feed contact surfaces in feed mills (0.82%), and finished feed (0.70%) compared with environmental samples collected from animal transport vehicles and contact surfaces at a company-owned market pig transfer station (4.13%). In contrast to these feed mill sampling surveys, Elijah et al. [40] used qPCR analysis to evaluate the distribution of ASFV within a feed mill after manufacturing experimentally inoculated feed and observed detectable ASFV DNA in all feed and non-feed contact and transition zones, ranging from 38% to 100% depending on the surface. Similarly, Schumacher et al. [98] used qPCR analysis of environmental swabs collected from feed and non-feed contact surfaces in a pilot-scale feed manufacturing facility involving feed that was experimentally inoculated with PEDV. Positive PCR results were obtained for all samples from all feed contact surfaces and nearly all non-feed contact surfaces. Comparing results between real-world sampling surveys with those from experimental studies is a reminder that one should not assume that experimental results from feed mill contamination studies are representative of real-world surveys of feed mill contamination.

Biosecurity and mitigation strategies to reduce the risk of bacterial and viral pathogen contamination in feed mills have been evaluated and summarized based on a limited number of studies [32,99]. Feed mill decontamination strategies that have been evaluated include use of extended holding times during storage, mechanical reduction in virus concentration, chemical cleaning and sanitizing surfaces, thermal processing and irradiation, and the addition of various feed additives and acidifiers to contaminated feed for various viruses. Wu et al. [100] conducted detailed sampling and evaluation of potential routes of introducing PEDV into a Chinese swine production system and reported that excluding high-risk ingredients in diets, increased thermal processing during pelleting, and a 7-day feed quarantine from delivery to consumption decreased the prevalence of PEDV-related disease after these practices were implemented. Nearly all of the studies conducted have evaluated PEDV decontamination strategies, which may not be applicable to other viruses, because viruses vary in their structural and functional characteristics and often respond differently to thermal and chemical mitigants. Therefore, generalizing mitigation responses among viruses and feed ingredients should be avoided. 

One of the simplest methods for decontaminating virus-contaminated feed and feed ingredients is to store potentially contaminated batches for an extended period of time in a heated warehouse or feed mill. Several government protocols have been developed and implemented in Canada [101], the European Union [102], and Australia [103] that require strict guidelines for imported ingredients from high-risk countries, which include extended storage time, to minimize the likelihood of introducing a foreign animal disease. Voluntary biosecurity protocols that include extended storage in heated warehouses have also been developed and implemented for imported feed ingredients in the U.S. [53]. However, as previously discussed, although heat exposure accelerates virus inactivation, it also accelerates loss in the nutritional value of feed ingredients including vitamins [104], amino acids [105,106], and lipids [107], as well as the biological activity of feed additives such as enzymes [108] and probiotics [109]. Furthermore, mold, mycotoxin, and bacterial growth may occur, depending on the moisture content and water activity of ingredients.

Mechanical reduction of PEDV concentration in experimentally contaminated feed manufacturing facilities has been evaluated using batch sequencing [110] and flushing mixers and equipment with rice hulls containing FMPA or MCFA blends [89]. Although these methods were somewhat effective in reducing PEDV nucleic acids, neither practice was completely effective in eliminating virus. Similarly, Elijah et al. [111] evaluated the effect of batch sequencing as a decontamination technique in a pilot-scale feed mill experimentally contaminated with ASFV and found that concentrations decreased sequentially with increasing batches, but virus was still detectable after the fourth batch. Therefore, other mitigation measures beyond batch sequencing and flushing using chemical mitigants are needed to eliminate viruses from contaminated feed manufacturing systems.

Because all viruses have some sensitivity to heat exposure, the heat provided during the conditioning and pelleting process in feed mills can be effective in reducing or eliminating the infectivity of swine viruses. Cochrane et al. [78] conducted studies to determine if the time and temperature applied to PEDV-contaminated feed during the pelleting process was capable of sufficient virus inactivation to prevent a PEDV infection when fed to pigs. Using different combinations of conditioning temperature and retention times to pellet feed inoculated with a low or high dose of PEDV resulted in no infections when fed to pigs compared with feeding the unprocessed feed containing inoculated virus. They also showed that feed being processed at 54 °C or more, using a 30 s retention time, prevented PEDV infections when fed to pigs compared with feed pelleted at 38 °C or 46 °C. 

The effectiveness of chemical cleaning and sanitizing feed mill equipment and surfaces to reduce PEDV concentration has also been evaluated. Huss et al. [112] applied a quaternary ammonium-glutaraldehyde blend cleaner, a sodium hypochlorite sanitizing solution, or heated a feed manufacturing facility up to 60 °C for 48 h to measure PEDV nucleic acid concentrations on surfaces. All of these methods were somewhat effective in reducing PEDV nucleic acids, but none of them were completely effective in eliminating the virus. In summary, the limited effectiveness of decontamination strategies in feed mills using common decontamination strategies in the limited number of studies conducted emphasizes the need for adhering to strict feed supply chain biosecurity protocols for prevention, because once a feed mill becomes contaminated with viruses, it is difficult to totally eliminate them.

## 10. Effects of Functional Ingredients, Nutrients, and Commercial Feed Additives during a Viral Health Challenge

Several plant extracts contain compounds with antiviral properties, including flavonoids, alkaloids, phenolic acids, terpenes, coumarins, lignans, and proteins [113]. Of these compounds, most of the previous research has focused on the effects of dietary flavonoids (i.e., isoflavones) during viral challenges in growing pigs. 

### 10.1. Soy Isoflavones and PRRSV Challenges

Isoflavones are flavonoid compounds that have potent antiviral properties against a wide variety of viruses, including enveloped and nonenveloped, single-stranded and double stranded, and RNA and DNA viruses [114]. Soybean products, including soybean meal, soy protein concentrate, and soy protein isolate, which are commonly used in swine diets, are rich sources of isoflavones (i.e., genistein, daidzein, glycitein) that have anti-inflammatory, antioxidative, and antiviral properties. Genistein has been the most extensively studied and has been shown to reduce infectivity of many types of human and animal viruses at physiological and supraphysiological concentrations [114]. Soybean meal is added to swine diets at higher dietary inclusion rates than soy protein concentrate and soy protein isolate, and contains greater concentrations of total isoflavones (2096 mg/kg) compared with soy protein isolate (911 mg/kg) and soybean protein concentrate (115 mg/kg) [115]. Although much is known about the biological properties of flavonoids, their antiviral properties have not been completely characterized [114]. In addition, soy products also contain saponins, which are involved in anti-inflammatory pathways, immunomodulatory activities that enhance passive immunity, and increase immune responses from vaccines [115]. However, less is known of their antiviral effects than isoflavones. 

Of all swine viruses, PRRSV is the only virus that has received research attention relative to the dietary benefits of soy isoflavones. Several studies have shown consistent growth performance and health benefits from feeding diets with high amounts of soybean meal to pigs infected with PRRSV. Results from initial studies showed that dietary daidzein provided less improvement in the growth of weaned pigs during a PRRSV challenge [116] than genistein, which also improved systemic virus elimination of PRRSV-infected weaned pigs [117]. Furthermore, greater improvements in growth and immune responses have been observed when PRRSV-challenged pigs were fed high amounts of soybean meal (and isoflavones) compared with lower dietary inclusion rates [117,118]. The mechanisms of these isoflavone responses involve reducing viral replication and infectivity, expression of pro-oxidative signaling pathways, and the production of pro-inflammatory and anti-inflammatory cytokines in the immune system [115]. However, subsequent studies evaluating dietary isoflavone supplementation from soybean meal showed no improvement in the growth performance of nursery pigs [119] or inconsistent improvements in the growth performance of wean-to-finish pigs [120] infected with PRRSV, but a more robust immune response to PRRSV was observed in both studies. Feeding soy isoflavones reduced mortality by 50% in PRRSV-infected pigs [120], but this response appeared to not be associated with alterations in gut microbiome [121]. Although the mechanisms of these immune responses have not been determined, there is substantial scientific evidence that indicates that isoflavones in soybean meal are effective in reducing the detrimental health and growth performance effects of PRRSV infection in pigs. Because of the antiviral activity over a wide range of viruses, more research is needed to determine if these beneficial effects can be achieved when feeding soy isoflavones to pigs challenged with CSFV, FMDV, ASFV, and PEDV.

### 10.2. Animal Plasma

Spray-dried animal plasma contains many functional compounds, including immunoglobulins, albumin, fibrogen, lipids, growth factors, biologically active peptides, transferrin, enzymes, and hormones [80], that play a positive role in the immune system [122], especially in weaned pigs undergoing a disease challenge [123]. Blázquez et al. [79] collected unprocessed liquid-porcine-plasma-contaminated ASFV from the blood of infected pigs, blended it with feed to achieve an infectious dose of 10^4^ or 10^5^ TCID_50_, and fed the contaminated feed for 14 consecutive days to determine if it would cause infection in naïve weaned pigs in two separate experiments. None of the pigs in either experiment became infected, indicating that either the minimum infective dose of ASFV is greater than 10^5.0^ log TCID_50_/pig or that liquid porcine plasma has significant functional properties that may reduce the infectious capability of ASFV. Additional evidence of the functional benefits of feeding spray-dried animal plasma to weaned pigs was observed in a study conducted by Crenshaw et al. [124], which showed that feeding diets containing spray-dried bovine plasma to pigs infected with PRRSV resulted in greater final body weight and reduced mortality compared with pigs fed diets with other specialty proteins and feed additives. 

Feeding spray-dried animal plasma to weaned pigs also appears to enhance immune response to vaccines. Weaned pigs fed a starter diet containing SDPP and vaccinated for porcine circovirus type 2 and Mycoplasma hyopneumoniae supported the best long-term benefits in survival to market and carcass weight [125]. More recently, Pujols et al. [126] determined if feeding a diet containing 8% SDPP would enhance the efficacy of a candidate ASFV vaccine when naïve pigs were directly exposed to pigs infected with ASFV Georgia 2007/01. Results of this study showed that no virus was detected in any organ of pigs fed SDPP, and pigs had lower viral load in blood, nasal, and rectal secretions after the ASFV challenge, indicating improvement in vaccine efficacy and health under ASFV challenge conditions. Another study showed that feeding diets containing 8% SDPP to weaned pigs reduced ASFV transmission and disease progression by enhancing ASFV-specific T-cell responses [127]. These results, combined with the demonstrated inactivation capabilities of ASFV and other swine viruses in porcine plasma during the spray-drying process, indicate that animal plasma is part of the solution for disease prevention rather than a potential risk factor.

### 10.3. Monoglycerides and Medium Chain Fatty Acids

Monoglycerides and MCFA have become one of the most important types of antiviral feed additives for use in swine diets, and their molecular properties and biological functions have been reviewed and summarized [86]. Medium-chain fatty acids are a group of saturated fatty acids with 6 to 12 hydrocarbons in their structure, and along with monoglycerides, have been shown to inactivate enveloped viruses [128]. Virus inactivation from MCFA is caused by disruption of the bilayer-lipid membranes in the viral envelope that protects nucleic acids by forming micelles, while monoglycerides form micelles at lower concentrations, suggesting greater potency than MCFAs [129,130]. In addition, PEDV is a single-stranded, enveloped RNA virus that is susceptible to inactivation by MCFA and monoglycerol. Phillips et al. [96] added a proprietary monoglyceride blend or a MCFA blend to feed inoculated with PEDV and fed these diets to nursery pigs for 20 days and observed no PEDV infections when diets contained either feed additive compared with pigs fed untreated diets. 

Hanczakowska [131] summarized the results of several studies showing the positive growth performance effects from feeding swine diets supplemented with MCFA. These responses were confirmed in a study by Gebhardt et al. [93], which showed that feeding diets containing increasing concentrations of an MCFA blend (1:1:1 of C6:0, C8:0, and C10:0) resulted in a linear increase in growth rate and gain efficiency compared with feeding non-supplemented diets. These researchers also inferred that feed containing the MCFA blend retained PEDV mitigation activity after a 40-day storage period, but they did not evaluate virus infectivity using virus isolation or a pig bioassay.

### 10.4. Potential Antiviral Components for Use in Swine Feed

#### 10.4.1. Plant Extracts

Most of the research conducted to study the antiviral effects of flavonoids, alkaloids, phenolic acids, terpenes, and coumarins in plant extracts has involved either human coronaviruses or influenza viruses [113]. Positive results from human coronavirus studies imply that some of these compounds may be effective against PEDV, PDCoV, and TGEV in pigs. Some compounds in the natural extracts of medicinal herbs and plants have been shown to inhibit viral replication of coronaviruses [132]. In addition to soy isoflavones and saponins, several other naturally occurring plant flavonoids have antiviral activity against ASFV in vitro by targeting different stages of the viral life cycle [133,134]. 

##### Other Flavonoids

Effective treatments for pigs infected with ASFV are desperately needed because no effective and safe vaccines are available to prevent ASFV infection in pigs. Studies have shown that nucleoside analogues, interferons, specific flavonoids, a limited number of antibiotics, and small interfering RNA molecules inhibit ASFV replication by either acting directly as antiviral compounds or specifically providing certain antiviral effects in the host [135]. Several in vitro studies have screened and tested flavonoid compounds to determine their potency and antiviral activity against ASFV. Hakobyan et al. [133] evaluated the antiviral effect of five flavonoids on the replication of ASFV in Vero cells and reported that apigenin had the greatest dose-dependent antiviral effect on ASFV. However, because apigenin is insoluble in polar solvents and occurs in derivative forms in plants, Hakobyan et al. [136] also screened several commercially available apigenin derivatives and showed that genkwanin had the most potent antiviral activity against a highly virulent field strain of ASFV. Arabyan et al. [134] showed that non-cytotoxic concentrations of genistein reduced ASFV infection in Vero cells and porcine macrophages. In a subsequent study, Arabyan et al. [137] screened 90 flavonoid compounds using a cell-based colorimetric assay and identified 9 flavonoids that had more than 40% inhibition of ASFV without any cell monolayer damage, which included 7,8-benzoflavone, calycosin, diosmin, isosinensetin, kaempferol, khellin, maackiain, sakuranetin, and sinensetin. However, kaempherol was the most potent and provided a dose-dependent response against a highly virulent ASFV isolate, which makes it a promising anti-viral candidate against ASFV. Further research is needed to evaluate and compare the potential antiviral effects of genistein, genkwanin, and kaempferol when added to swine diets and fed to ASFV-infected pigs. 

##### Fluoroquinolones

Fluoroquinolones are a class of antibiotics that are approved for use in treating certain types of bacterial infections but have also been shown to exhibit potent antiviral properties. These antibiotics trap DNA gyrases and topoisomerase IV on DNA and promote the formation of drug enzyme-DNA cleavage complexes that cause disruption of DNA replication, leading to mechanisms resulting in cell death [138]. Modifications in the molecular structure of fluoroquinolones have been shown to provide antiviral properties against RNA and DNA viruses. Phylogenetic studies have suggested that antibacterial topoisomerase inhibitors such as fluoroquinolones may interfere with ASFV replication [139,140]. Therefore, Mottola et al. [138] conducted an in vitro study to screen 30 fluoroquinolones for antiviral activity against ASFV. These researchers identified six fluoroquinolones and some combinations provided a severe reduction in the cytopathic effects on ASFV-infected Vero cells in the early stage of infection followed by non-detectable ASFV genome and infectivity after 7 days, which suggests that selected fluoroquinolones or their combinations may be effective antiviral treatments for ASFV when fed to pigs.

#### 10.4.2. Salts

Limited studies have evaluated the effects of various salts on swine virus inactivation. One study involved experimentally infecting pigs with CSFV and ASFV, euthanizing them during the acute phase of disease when viremia was greatest, and collecting small and large intestine samples for incubation with either sodium chloride or a salt mixture (86.5% sodium chloride, 2.8% trisodium phosphate, and 10.7% disodium hydrogen phosphate) at various temperatures up to 20 °C for multiple times up to 60 days [141]. Both sodium chloride and the salt mixture were effective in accelerating inactivation of CSFV and ASFV in a temperature-dependent manner. Other studies have shown some inactivation of PDCoV, PEDV, and TGEV in some feed ingredients and complete feeds with the addition of sodium chloride [58,87]. Sodium chloride is an attractive mitigant because it is inexpensive and commonly supplemented in swine diets to meet nutritional requirements. However, more dose titration studies with sodium chloride and other salt mixtures are needed to determine their feasibility as effective viral mitigants and the role of dietary cation and anion concentrations on the inactivation of various types of swine viruses.

#### 10.4.3. Copper and Zinc

Metals such as copper and zinc possess several properties including redox, photocatalytic, and structural stability, along with antibacterial and antiviral properties [142], that suggest that their use as antiviral agents in virus-contaminated swine feeds is worth exploring. Copper ions, alone or in copper complexes, have potent antibacterial and antiviral activity [143]. Feeding pharmacological concentrations of copper has been a common practice for many decades as a low-cost and effective way of consistently improving growth performance and reducing post-weaning diarrhea in weaned pigs [144]. Similarly, zinc has been shown to have antimicrobial as well as direct inhibitory effects on several viruses [145]. When pharmacological levels of dietary Zn, in the form of zinc oxide, are fed to weaned pigs, it has been shown to be effective in controlling non-specific post-weaning diarrhea and promoting growth [146]. However, dietary concentrations of copper and zinc are restricted and far below pharmacological-use levels in several countries in the European Union, which may also prevent their use as antiviral agents in contaminated feed ingredients. Read et al. [147] summarized numerous studies and reported that zinc not only has direct antiviral properties, but it also plays a critical role in innate and acquired antiviral responses. In addition, zinc is a component of several viral enzymes, proteases, and polymerases, which are involved in virus replication and dissemination. Wei et al. [145] compared the antiviral effects of zinc chloride and zinc sulfate when applied to swine testicle cells infected with TGEV and showed that although these zinc salts had no effect on TGEV-cell binding, antiviral effects were observed through the inhibition of virus penetration or exit or the intracellular phase of the TGEV life cycle. Although the chemical structures of metals such as copper and zinc affect their ability to inactivate viruses, their redox capability appears to be a key chemical component affecting antiviral activity [142]. Furthermore, the use of copper and zinc nanoparticles may not only provide direct antiviral activity but may also provide therapeutic effects on animals infected with viruses [142]. Nanoparticles of zinc provide the advantage of greater growth-promoting, antibacterial, and immune responses at lower doses compared with conventional sources [148]. Therefore, nanoparticles of copper and zinc should be evaluated for their potential benefits as chemical mitigants to inactivate swine viruses in feed, as well as their potential role in alleviating adverse health effects during viral disease challenges. 

### 10.5. Commercially Available Chemical Mitigants

The goal of feed supply chain biosecurity programs is to deliver complete feeds to swine farms that are devoid of disease-causing pathogens. However, if viral pathogen contamination is suspected in complete feed delivered at the farm level, the addition of several commercially available feed additives to swine diets may improve the health and growth performance of pigs fed contaminated feed and undergoing a disease challenge. Many antiviral commercial feed additives containing various combinations of MCFA blends, glycerol monolaurate, organic acids, essential oils, and various other compounds have been developed, approved for use, and commercially available in some countries.

Fifteen commercially available feed additives were evaluated when added to nursery feed contaminated with PRRSV, SVV-A, and PEDV to determine their effectiveness in improving the health and growth performance of pigs [88]. A series of feeding trials were conducted using weaned pigs (5 to 8 weeks of age) that were fed non-contaminated feed for a 10-day pre-challenge period followed by a 15-day post-challenge period to confirm viral infection and determine clinical scores for diarrhea, lameness, and dyspnea, as well as growth and mortality rates, when diets containing various antiviral commercial feed additives were fed. The majority (14 of 15) feed additives evaluated in these trials improved pig health and growth rate during the 15-day post-challenge period compared with pigs consuming virus-contaminated feed without additives. Furthermore, pigs fed virus-contaminated feed containing 10 of these 15 feed additives had no signs of clinical disease, very low mortality (≤1%), and greater ADG compared with control groups. Only one of the feed additive products tested was ineffective for improving the health and growth rate of weaned pigs fed virus contaminated feed. 

Two of the feed additives evaluated in this study [88] were also evaluated by Stenfeldt et al. [55] to determine their effectiveness in nursery pig diets contaminated with a strain of FMDV. These researchers added 10^8.3^ TCID_50_ FMDV A24 (greater than the previously determined minimum infectious dose) to feed with or without SalCurb or Guardian feed additives 24 h prior to feeding and observed no clinical signs or positive antemortem samples for pigs fed either feed additive treatment, except for one pig fed Guardian that was considered subclinically infected. 

Beyond the results reported in these studies, it is unknown if these additives are effective in improving health and growth rate in swine diets contaminated with ASFV or other swine viruses. However, these results suggest that several commercially available feed additives may be effective as a last defense against a biosecurity breach to minimize adverse health effects from PRRSV-, SVV-A-, and PEDV-contaminated swine feed.

## 11. Future Considerations for Risk Assessment Model Development

Quantitative risk assessment of virus transmission through various feed ingredient supply chains is greatly needed because of high uncertainty due to the lack of a global monitoring and surveillance system. Unfortunately, most publications in the literature do not provide sufficient details and data to enable the development of quantitative risk assessment models. Many experiments evaluate only a single temperature, sampling time, or measure of virus inactivation rather than evaluating a comprehensive set of conditions. Furthermore, very few studies have reported D-values or z-values for virus inactivation kinetics that allow for comparison of results among studies. Accurate estimates for kinetic parameters of virus inactivation require multiple time and temperature conditions. Typically, virus inactivation curves follow non-linear patterns, which are best modelled with at least 4–5 observations distributed along the expected range in temperatures and sampling times in virus inactivation models. A limited number of replications per timepoint or temperature is another common problem with the data provided in the published literature. Many researchers fail to recognize the appropriate experimental unit and the number of experimental units associated with the error term measurement of the virus inactivation. Pilot experiments can be a useful approach for collecting preliminary data on the variability of conditions associated with virus inactivation to determine optimal subsequent experimental design. In the predictive modeling of pathogen inactivation, there are sources of uncertainty and variation in observed predictions that are introduced due to unknown effects of independent factors. Another source of variation is actual variability in the process input range. Because this source of variation is known, it should be estimated using sensitivity analyses. In summary, researchers are encouraged to consider these key deficiencies when reporting virus inactivation data from various mitigation strategies in feed ingredients in future studies to enable subsequent quantitative risk assessment determinations. 

## 12. Conclusions

Compared with other virus transmission routes, feed ingredients and complete feed appear to be less likely contributors toward disease transmission, but because there is no monitoring and surveillance system, there is high uncertainty of the extent of swine virus contamination in global feed supply chains. Biosecurity protocols need to be developed and implemented to improve our ability to prevent virus contamination and transmission through the production, processing, storage, and transportation of swine feed. Key components of feed biosecurity protocols should also include effective mitigation practices such as extended storage times, thermal and irradiation processing, and chemical mitigants to provide inactivation of viable swine viruses if they are present. Several types of functional feed ingredients, nutrients, and feed additives that have antiviral properties need to be further evaluated for their ability to inactivate swine viruses of concern in various types of feed ingredients. 

Unfortunately, there are numerous challenges that must be overcome to improve our understanding and ability to accurately predict whether feed contaminated with swine viruses is capable of causing an infection, including limitations of current analytical methods for measuring virus inactivation, viability, and infectivity in feed. In addition, the use of the food safety objective and performance objective need to be developed for risk assessment of virus survival in feed ingredients. Improving data quality and quantity when reporting results in scientific publications is needed to provide sufficient detail to allow for developing risk assessment models and calculating D-values and z-values for virus inactivation kinetics that allow for comparison of results among studies.

## Figures and Tables

**Table 2 animals-13-02375-t002:** Summary of estimate of minimum infectious doses of swine viruses through consumption of inoculated feed or direct oral inoculation in pigs.

Virus	Minimum Infectious Dose	Observations	Reference
African swine fever virus	10^4^	5	[61]
	>10^5.0^ TCID_50_/pig	8	[68]
Classical swine fever virus	10^4.2^ TCID_50_ to 10^5.5^ TCID_50_ depending on strain	6	[66]
Foot and mouth disease virus	10^6.2^ TCID_50_ to 10^7^ TCID_50_ depending on strain	4	[55]
	10^5.5^ TCID_50_/mL	2	[69]
Porcine epidemic diarrhea virus	10^5.6^ TCID_50_/g	3	[70]
Porcine reproductive and respiratory syndrome virus	10^5.3^ TCID_50_	36	[71]
Seneca Valley A virus	10^3.1^ TCID_50_/mL for neonates, 10^2.5^ TCID_50_/mL market-weight pigs	4	[67]

**Table 3 animals-13-02375-t003:** Comparison of differences between using a food safety objective versus a performance objective as potential approaches for assessing swine virus risk assessment in feed ingredients and complete feeds.

Item	Food Safety Objective (FSO)	Performance Objective (PO)
Defined as:	Safe microbiological level of frequency of intake of a given feed ingredient or complete feed at the time of consumption	Safe microbiological level in a given feed ingredient or complete feed at the time of production and before consumption
Interpreted as:	Maximum concentration of a microorganism or hazard allowed at the time of consumption	Maximum concentration of a microorganism or hazard allowed at a specified step in the processing chain
Applied to:	The FSO is related to the contamination of the raw material and inactivation achieved during the individual or multiple control steps	The PO is related to the contamination of the raw material and inactivation achieved during the individual or multiple control steps, and it can also be applied to feed safety
Conditions for use:	Requires establishing the size of the population to protect, frequency of consumption, and level of exposure	Requires establishing a quantity of product to deem as the PO, such as batch of product processed
Application in swine diets:	The FSO concept can be applied to feed safety involving swine viruses to protect the health status of an entire pig farm but has not yet been established	A PO level related to the presence of swine viruses has not been established for any feed ingredient

**Table 4 animals-13-02375-t004:** Feed manufacturing processes that reduce the concentration or inactivate porcine epidemic diarrhea virus in feed ingredients or complete feed (adapted from [75]).

Process	Range in Temperature and Time	Results
Pelleting complete feed	68–95 °C for 9–240 s and 14% to 18% final moisture	2 log reduction of PEDV in feed at >54 °C
Extrusion of soybean meal and complete feed	80–200 °C for 5–10 s and 20–30% final moisture	Temperature and time likely to reduce PEDV concentration, but validation study is needed to quantify virus reduction
Expansion of various ingredients and complete feeds	90–150 °C for 1–4 s and 10–80 bar pressure	Temperature and time likely to reduce PEDV concentration, but validation study is needed to quantify virus reduction
Desolventizing and toasting soybean meal	Up to 120 °C for 10–20 min	Temperature and time likely to reduce PEDV concentration, but validation study is needed to quantify virus reduction
Rendering of animal fats and protein by-products	115–145 °C for 40–90 min	3.7 to 21.9 log reduction of PEDV
Spray drying of animal plasma	Inlet air = 150–200 °C; outlet air = 80 °C for 20–90 s	4.2 log reduction at 80 °C
Steam flaking of grain	15 °C initial temperature increasing to 100 °C at 14% moisture	Temperature and time likely to reduce PEDV concentration, but validation study is needed to quantify virus reduction
Irradiation of various complete feeds and ingredients	Gamma rays, X-rays, and electron beams (FDA approved up to 50 kGy)	3 and 5 log reduction of PEDV after 50 and 86.25 kGy exposure, respectively
Extended storage of complete feeds and ingredients	Ambient air temperature > 18 °C for 2 weeks	3 to 5 log reduction of PEDV at 20 °C for 2 weeks

**Table 5 animals-13-02375-t005:** Summary of studies evaluating inactivation of swine viruses inoculated in feed ingredients and complete feed and subjected to various thermal processing conditions.

Virus	Matrix	Process Conditions	Detection Method	Initial Virus Concentration	Viral Reduction	Reference
African swine fever virus	Porcine plasma	Lab-scale spray drying with inlet air of 200 °C, outlet air of 80 °C and drying time < 1 s	Titration assay using Vero cells	10^6.9^ TCID_50_/mL	4.11 log reduction after spray drying	[79]
	Porcine plasma	4, 21, or 48 °C; 7.5 or 10.2 pH; 0 or 92.6 mM H_2_O_2_; 1 to 90 min	Endpoint dilution assays using Vero cells	10^4.71^ TCID_50_/mL Exp. 1 10^4.62^ TCID_50_/mL Exp. 2 10^8.35^ TCID_50_/mL Exp. 3	3.35 to 4.17 log reduction when treated with 48 °C, pH 10.2, 20.6 or 102.9 mM H_2_O_2_ for 10 min	[84]
	Corn, wheat, barley, rye, peas, triticale	Lab-scale drying for 2 h at room temperature or drying for 2 h and heating for 1 h at 40, 45, 50, 55, 60, 65, 70, and 75 °C	Rt-PCR Haemadsorption test	20 g samples of each ingredient inoculated with 900 μL infectious blood with 10^6^ HAD_50_/mL	No viable virus was recovered after 2 h of drying at room temperature and after heat treatment at any temperature	[82]
	Corn, soybean meal, meat and bone meal	Lab-scale inoculation and incubation at 60, 70, 80, and 85 °C	Titration assay	1 g of each ingredient was added to 15 mL centrifuge tubes and 500 μL of ASFV suspension containing 10^5^ HAD_50_/mL was added	Heat resistance was not different among at 60, 70, 80, and 85 °C with D values ranging from 5.11–6.78, 2.19–3.01, 0.99–2.02, and 0.16–0.99 min, respectively	[83]
Classical swine fever virus	Porcine plasma	Lab-scale spray drying with inlet air of 200 °C, outlet air of 80 °C and drying time < 1 s	Titration assay using PK-15 cells	10^7.5^ TCID_50_/mL	5.78 log reduction after spray drying	[79]
Porcine epidemic diarrhea virus	Porcine plasma	Lab-scale spray drying with inlet air of 166 °C, outlet of 80 °C and drying time < 1 s	Rt-PCR Sequencing Pig bioassay	10^4.2^ TCID_50_/mL	4.2 log reduction after spray-drying and storage for 7 days at 4 °C	[77]
	Porcine plasma	Lab-scale spray drying with inlet air of 200 °C, outlet of 80 °C and drying time < 1 s	Microtiter assay using Vero cell monolayers	10^4.2^ TCID_50_/mL 10^5.1^ TCID_50_/g	4.2 log reduction after spray-drying and heating in water bath	[56]
	Complete feed	Oven incubation at 120 °C to 145 °C for up to 30 min	Microtiter assay using Vero cells	6.8 × 10^3^ TCID_50_/mL	D values ranged from 16.52 min at 120 °C to 1.30 min at 145 °C; 130 °C for >15 min caused 99.9% loss of virus infectivity	[76]
	Complete feed	Pelleting temperature of 68.3, 79.4, and 90.6 °C; conditioning times of 45, 90, or 180 s	rtPCR Pig bioassay	10^2^ TCID_50_/g or 10^4^ TCID_50_/g	No PEDV RNA was detected in fecal swabs or cecum contents 7 days after inoculation at either dose or any of the 9 processing combinations	[78]
	Complete feed	Pellet-conditioning temperatures of 37.8, 46.1, 54.4, 62.8, and 71.1 °C; conditioning times of 30 s	rtPCR Pig bioassay	10^4^ TCID_50_/g	All samples had detectable PEDV RNA, but only samples from 37.8 and 46.1 °C were infective	[78]
	Corn, soybean meal, DDGS *, SDPP **, blood meal, meat and bone meal, meat meal, vitamin-trace mineral premix	Lab-scale water bath incubation at 60, 70, 80, and 90 °C for 0, 5, 10, 15, or 30 min	Microtiter assay using Vero cells	3.2 × 10^4^ TCID_50_/mL	3.9 log reduction of all ingredients at 90 °C for 30 min, but no differences in virus survival among feed ingredients regardless of time and temperature. Different combinations of time and temperature resulted in a 3 to 4 log reduction in virus in all ingredients	[58]
Porcine reproductive and respiratory syndrome virus	Bovine plasma	Pilot-scale spray drying with inlet air at 240 °C and outlet of 90 °C for 0.41 s	MARC cell culture using indirect fluorescent antibody procedure	10^3.5^ TCID_50_/mL to 10^4.0^ TCID_50_/mL	No virus infectivity was detected after spray drying	[81]

* DDGS = corn distillers dried grains with solubles; ** SDPP = spray-dried porcine plasma.

**Table 6 animals-13-02375-t006:** Summary of studies evaluating inactivation of swine viruses inoculated in feed ingredients and treated with various chemical mitigants.

Virus	Matrix	Mitigants Evaluated *	Inclusion Rates	Detection Method	Experimental Conditions	Results	Reference
African swine fever virus	Conventional soybean meal, organic soybean meal, soy oil cake, choline chloride, moist dog food, moist cat food, dry dog food, pork sausage casings, complete feed	FMPA, MCFA	0.03 to 2.0%	Cell culture TCID_50_ using Vero cells; PCR; virus isolation; pig bioassay	Average temperature 12.3 °C at 74% relative humidity for 30 days in shipping model	Dose-dependent virus inactivation with 0.35% FMPA and 0.7% MCFA required to reduce virus titers below level of detection in cell culture; all treated feed samples had detectable nucleic acids on day 1, 8, 17, and 30 of shipping model conditions but virus isolation showed no detectable virus at 30 days; only 1 sample of organic soybean meal and 1 sample of dry dog food of the 36 matrices tested resulted in ASFV infection in bioassay	[85]
	Complete feed	MCFA blend, GML	0.25, 0.50, 1.0, and 2.0%	Cell culture TCID_50_ using Vero cells, Rt-PCR, ELISA	Feed stored for 30 min or 24 h at room temperature	Virus titers in cell culture decreased by MCFA and GML; GML was more potent than MCFA at lower doses and one or more antiviral mechanisms; dose-dependent effect by GML within 30 min; reduced infectivity by GML at ≥1.0%; no effect on viral DNA	[86]
Foot and mouth disease virus	Pelleted complete feed, DDGS **, soybean meal	FMPA, MCFA, lactic acid-based acidifier	FMPA (0.33%), MCFA (1%), lactic acid product (0.44%)	Cell culture TCID_50_ using LFBK-αvβ6 cells, virus viability, virus isolation, calculated half-life	Viability of 1 FMDV strain tested at 1 h and 1, 3, 7, 14, 21, and 37 days post inoculation at 4 °C or 20 °C	FMPA treatment reduced virus titers below detection by 1 day at 20 °C and 3 days at 4 °C with infectious virus isolated at 7 days at 20 °C and 37 days at 4 °C; lactic-acid-based additive reduced titers below detection by 3 days at both temperatures, but infectious virus was isolated up to 14 days at 20 °C and 37 days at 4 °C; MCFA treatment had no effect on reducing virus below detection up to 37 days at 4 °C, but was below detection by 14 days at 20 °C, and infectious virus was isolated at 21 days; FMPA reduced infectivity of complete feed within 24 h at 20 °C, and lactic-acid-based product also reduced infectivity despite questionable reduction virus viability in vitro	[55]
Porcine delta corona virus	Complete feed	Commercial organic acids, HMTBa blend with organic acids, acidifiers, sucrose, sodium chloride	Exp 1.—recommended doses of 10 to 150 mg or 46 to 56 μL; Exp. 2—2 times recommended doses of 20 to 300 mg or 92 to 112 μL	Cell culture TCID_50_ using swine testicular cells; inactivation kinetics using D values based on Weibull model	Feed stored at 25 °C for 35 days and sampled at 0, 7, 14, 21, 28, and 35 days in Exp. 1 and 0, 1, 3, 7 and 10 days in Exp. 2.	No differences in virus inactivation at recommended doses; 2 times the recommended doses were effective for inactivation except for one product; products with phosphoric acid, citric acid, fumaric acid were most effective; none completely inactivated virus by 10 days post-inoculation	[87]
Porcine reproductive and respiratory syndrome virus, porcine epidemic diarrhea virus, and Seneca Valley A virus	Complete feed	FMPA, organic acids, benzoic acid, HMTBa, SCFA, MCFA, LCFA, GML, essential oils, prebiotic fiber, bacterial fermentation products	0.1 to 3.0%	Feed and oral fluid samples collected on day 0, 6, 15 post-challenge; necropsy on subset of pigs on day 15 post-challenge; clinical signs, growth performance, and mortality were evaluated	Feed inoculated with a block of ice containing equal concentrations of PRRSV, PEDV, and SVV-A on day 0 and 6 of each 25-day experiment (10-day pre-challenge and 15-day post-challenge)	14 of the 15 commercial feed additive products improved growth rate, reduced clinical signs and infection levels, while feeding diets with 10 of the 15 additives resulted in no signs of clinical disease and ≤1% mortality compared with feeding control diets with no additives	[88]
Porcine epidemic diarrhea virus	Complete feed, DDGS, meat and bone meal, soybean meal, SDPP ***, spray-dried red blood cells, choice white grease, soybean oil, L-lysine HCl, DL-methionine, L-threonine, choline chloride, limestone, vitamin-trace mineral premixes	FMPA	0.33%	PCR, virus isolation, swine bioassay	320 feed ingredient samples stored under winter conditions (−9 °C to −18 °C) for 30 days and sampled on days 1, 7, 14, and 30 post-inoculation	Viable virus was detected by virus isolation or swine bioassay on days 1, 7, 14, and 30 post-inoculation in soybean meal, DDGS, meat and bone meal, spray-dried red blood cells, L-lysine HCl, DL-methionine, choice white grease, choline chloride, and complete feed, and at 7 days post-inoculation in limestone and 14 days post-inoculation in L-threonine; treatment with FMPA was effective for preventing clinical signs and positive PCR tests of the small intestine in all ingredients except choline chloride and choice white grease	[22]
	Rice hulls	FMPA, MCFA blend	0.33 FMPA 2% MCFA or 10% MCFA	PCR, swine bioassay	Untreated and treated rice hulls stored in double-lined bags for 48 h at 21 °C until initiation of flush in laboratory scale mixers; inoculation with virus prior to initiating flush	Flushing with 10% MCFA treated rice hulls resulted in no detectable virus RNA; 2 of 6 samples treated with 2% MCFA and 1 of 6 samples treated with 0.33% FMPA had detectable virus RNA; dust collected after mixing virus-contaminated feed in a production-scale mixer had detectable virus RNA that was infectious; treating rice hull flush with 10% MCFA or 0.33% FMPA reduced virus RNA after manufacturing PEDV-contaminated feed	[89]
	Organic soybeans, organic soybean meal, conventional soybeans, conventional soybean meal, L-lysine HCl, DL-methionine, L-tryptophan, vitamin A, vitamin D, vitamin E, choline chloride, rice hulls, corn cobs, tetracycline, complete feed	FMPA, MCFA	0.33% FMPA, 2.0% MCFA	PCR, virus isolation, swine bioassay	Range in temperature was 3.9 to 10 °C and relative humidity was 26 to 94% during the 37-day trans-Pacific shipping simulation study period. PEDV-inoculated feed was fed to PEDV-naïve pigs for 14 days to observe clinical signs of infection	Addition of FMPA reduced virus RNA, but 2.0% MCFA had no effect after 37 days; all FMPA- and MCFA-treated samples were negative for virus isolation across all batches; all pigs administered FMPA- and MCFA-treated ingredients were non-infectious and clinically normal throughout the testing period	[90]
	Complete feed	FMPA	0.32%	PCR, immunohistochemistry of gastrointestinal tracts, swine bioassay	PEDV-inoculated feed with or without FMPA was fed to PEDV-naïve pigs for 14 days to observe clinical signs of infection	FMPA prevented infection and clinical disease in PEDV-naïve pigs	[91]
	Complete feed	FMPA MCFA	0.3% FMPA, 0.125 to 0.66% of several individual MCFA, 1% MCFA blend	Rt-PCR, swine bioassay	4 experiments evaluated the addition of FMPA and varying inclusion rates of MCFA	All concentrations of MCFA were effective in reducing detectable PEDV RNA; all pigs had negative fecal swabs and Ct > 36 for virus when administered feed treated with FMPA, 0.5% MCFA blend, and 0.3% C8 MCFA	[92]
		MCFA blend, individual C6:0, C8:0, and C10:0 MCFA	0.25, 0.5, 1.0, and 1.5% MCFA blend; 0.5% C6:0, C8:0, or C10:0	Rt-PCR, swine bioassay	Various amounts of MCFA were added to experimental diets and stored for 40 days at 18.3 to 33.1 °C and average relative humidity of 90% prior to inoculating with PEDV virus and fed to pigs during a 35-day feeding period; feed samples were analyzed on day 0 and 3 post-inoculation for RNA	Addition of increasing dietary levels of MCFA blend and 0.5% of C6:0, C8:0, and C10:0 improved growth performance of pigs and provided residual mitigation activity against PEDV	[93]
	Complete feed	FMPA, MCFA blend, MG blend	3.1 kg/t 10 kg/t 1.5, 2.5, 3.5 kg/t	Cell culture TCID_50_ using Vero-81 cells, swine bioassay	Feed was inoculated using an ice block containing 10^5^ TCID_50_/mL of virus in feed bins and fed to pigs for 20 days; feed and oral fluid samples were collected on day 6 and 15 post-challenge, and rectal swabs and diarrhea prevalence were obtained on day 20 post-challenge	In vitro virus inactivation was FMPA = 2 log (99%) decrease in 24 h, MCFA = 99.79% decrease in 12 h, MG 1.5 = 2 log decrease in 24 h, MG 2.5 and 3.5 = 2 log decrease in 24 h; MCFA and MG blends reduced positive oral fluid and feed samples from feeders; rectal swabs were negative for all treatment groups	[94]
	Canola oil Choice white grease Coconut oil Palm kernel oil Soybean oil	FMPA MCFA blend, 0.66% C6:0, C8:0, C10:0, or C12:0	FMPA (0.33%); MCFA blend (1%); 0.66% C6:0, C8:0, C10:0, or C12:0; 1% of lipids	Swine bioassay	FMPA, MCFA blend, individual MCFA mitigants, and sources of fats and oils were added to diets	Addition of FMPA, 1% MCFA, 0.66% caproic, caprylic, and capric acid appeared to be effective in preventing infection, but not lauric acid or longer-carbon-chain lipid sources	[95]
	Complete feed	Lactic-acid-based acidifier	0.75, 1.0, 1.5%	Rt-PCR, virus isolation, swine bioassay	Feed samples containing increasing concentrations of mitigant were inoculated with PEDV and incubated for 24 h before testing; gnotobiotic pigs were orally inoculated with liquid supernatant	Feed samples containing lactic-acid-based acidifier were negative at all inclusion rates based on virus isolation; pigs inoculated with treated complete feed remained healthy, and rectal swabs were negative by Rt-PCR	[96]
	Complete feed	Benzoic acid, essential oils	0.5% benzoic acid 0.02% essential oil and combination in spray-dried plasma and swine gestation diet	Rt-PCR, swine bioassay	Feed samples analyzed for virus RNA on day 0, 1, 3, 7, 14, 21, and 42 and bioassay was conducted with 10-day-old pigs	The combination of benzoic acid and essential oil was most effective in reducing viral RNA; viral shedding was observed in spray-dried plasma and gestation diet treated with both feed additives on day 7 post-inoculation	[97]
	Complete feed	Organic acids, acidifiers, sucrose sodium chloride	0.25 to 1.5%	Cell culture TCID_50_ using Vero-81 cells; inactivation kinetics using D values based on Weibull model	Completed feed stored at 25 °C for 0, 1, 3, 5, 7, 14, and 21 days	All additives were effective in reducing virus survival; 2-hydroxy-4-methylthiobutanoic acid and a blend of phosphoric, fumaric, lactic, and citric acids provided the fastest inactivation of 0.81 and 3.28 days, respectively	[76]

* FMPA = aqueous formaldehyde and propionic acid; MCFA = medium-chain fatty acids; GML = glycerol monolaurate; HMTBa = methionine hydroxy analogue; SCFA = short-chain fatty acids; LCFA = long-chain fatty acids; MG = monoglyceride; ** DDGS = corn distillers dried grains with solubles; *** SDPP = spray-dried porcine plasma.

## Data Availability

All data summarized in the text and tables of this review article were obtained from previously published studies with appropriation attribution cited in text and tables and complete references provided in the Reference section.

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
