# Peer review of "Biosecurity and Mitigation Strategies to Control Swine Viruses in Feed Ingredients and Complete Feeds"

_animals, 2023, doi:10.3390/ani13142375_

Round 1
Reviewer 1 Report
Thorough, timely, and valuable review. Overall, well written. Excellent scientific contribution.
Limited discussion specifically made about feedstuffs of animal origin and if the risks differ for these feedstuffs as compared to vegetative feedstuffs. Lines 194-195 first mention; Table 1 presents feed products derived from animals; lines 651-653 and Section 7 presents greater discussion about these feed products specifically derived from animals but only in the context of heat and drying process. Are the risks at other times in the supply chain different for these feedstuffs?
Line 447 - first mention of 'water activity'. Definition of term would be beneficial. A definition could be made near lines 587-595, alternatively.
Inconsistent use of acronym SDPP and spray dried porcine plasma written out.
Line 980-981 Leading paragraph with comment about vaccine use misleads the reader, causing the reader to think the subsection may be about injectable mitigants and not consumables.
Lines 1085 to 1091 why use of product names here and not in other places in review?
Author Response
Thorough, timely, and valuable review. Overall, well written. Excellent scientific contribution.
Response: Thank you.
Limited discussion specifically made about feedstuffs of animal origin and if the risks differ for these feedstuffs as compared to vegetative feedstuffs. Lines 194-195 first mention; Table 1 presents feed products derived from animals; lines 651-653 and Section 7 presents greater discussion about these feed products specifically derived from animals but only in the context of heat and drying process. Are the risks at other times in the supply chain different for these feedstuffs?
Response: No quantitative risk assessment studies have been conducted to compare the relative risks of virus contamination in plant vs. animal origin feedstuffs, and limited data are available to do so [see ref 18 which describes the high uncertainty due to lack of data for determining risk of imported corn and soybean meal). Furthermore, different viruses seem to have different survival/inactivation patterns among different types of ingredients (see ref 23). Risk of contamination likely varies at different stage of ingredient supply chains but because there is no monitoring system in place, there are no data for making these types of comparisons. Therefore, it is not possible to accurately quantify and compare risk of virus contamination between plant vs. animal ingredients along supply chains.
Line 447 - first mention of 'water activity'. Definition of term would be beneficial. A definition could be made near lines 587-595, alternatively.
Response: Text was added to define water activity on lines589-592.
Inconsistent use of acronym SDPP and spray dried porcine plasma written out.
Response: SDPP was defined at first use in text and tables and replaced spray dried porcine plasma as suggested.
Line 980-981 Leading paragraph with comment about vaccine use misleads the reader, causing the reader to think the subsection may be about injectable mitigants and not consumables.
Response: Lines 987-989 have been revised as suggested.
Lines 1085 to 1091 why use of product names here and not in other places in review?
Response: We originally included a table with the names of commercial products that were evaluated and their chemical composition but have subsequently removed it. The product names in this sentence remained for brevity instead of describing the composition of both products (e.g., agueous formaldehyde and propionic acid for SalCurb).
Reviewer 2 Report
Comments, review of manuscript id: Animals-2482743.
Title: ”Biosecurity and Mitigation Strategies to Control Swine Viruses in Feed Ingredients and Complete Feeds”
The review manuscript gives an in-detail description on how to meet the risk for virus contamination and spreading of virus in feed components and complete feeds for pigs.
The manuscript is well written, and I have only some minor comments as shown below.
Comments.
In general, It might be an idea to include a list of the abbreviations used in the manuscript, then it will be easier for the reader to remember them.
Chapter 10.4.3. Copper and zinc., lines 1033-1060. You may consider mentioning that high levels of copper and zinc are restricted as feed additives in several countries. In Europe levels of zinc higher than the physiological requirements of animals is banned from June 2022.
Chapter 11, Future Considerations…,line 1100. A section in this sentence “… most publications the literature do not…” sounds strange. Is there a writing error here?
Reference 111, lines 1440-1441.Check if you should write the year of this publication “(2015)” in bold.
Author Response
In general, It might be an idea to include a list of the abbreviations used in the manuscript, then it will be easier for the reader to remember them.
Response: In our opinion, abbreviations were defined at first use in text and tables throughout the manuscript and adding a list of abbreviations and definitions at the beginning of the manuscript is redundant for an already lengthy manuscript.
Chapter 10.4.3. Copper and zinc., lines 1033-1060. You may consider mentioning that high levels of copper and zinc are restricted as feed additives in several countries. In Europe levels of zinc higher than the physiological requirements of animals is banned from June 2022.
Response: Text has been added on lines 1051-1054 as suggested.
Chapter 11, Future Considerations…,line 1100. A section in this sentence “… most publications the literature do not…” sounds strange. Is there a writing error here?
Response: Sentence has been revised as suggested on lines 1111-1112.
Reference 111, lines 1440-1441.Check if you should write the year of this publication “(2015)” in bold.
Response: Yes, thank you. It has been corrected.